# Evolution of a functionally intact but antigenically distinct DENV fusion loop

Rita M Meganck[1], Deanna Zhu[2], Stephanie Dong[2], Lisa J Snoderly-Foster[1], Yago R Dalben[1], Devina Thiono[3], Laura J White[3], Arivianda M DeSilva[3], Ralph S Baric[2], Longping V Tse[1]*

[1]Department of Molecular Microbiology and Immunology, Saint Louis University, Saint Louis, United States; [2]Department of Epidemiology, University of North Carolina at Chapel Hill, Chapel Hill, United States; [3]Department of Microbiology, University of North Carolina at Chapel Hill, Chapel Hill, United States

**Abstract** A hallmark of dengue virus (DENV) pathogenesis is the potential for antibody-dependent enhancement, which is associated with deadly DENV secondary infection, complicates the identification of correlates of protection, and negatively impacts the safety and efficacy of DENV vaccines. Antibody-dependent enhancement is linked to antibodies targeting the fusion loop (FL) motif of the envelope protein, which is completely conserved in mosquito-borne flaviviruses and required for viral entry and fusion. In the current study, we utilized saturation mutagenesis and directed evolution to engineer a functional variant with a mutated FL (D2-FL), which is not neutralized by FL-targeting monoclonal antibodies. The FL mutations were combined with our previously evolved prM cleavage site to create a mature version of D2-FL (D2-FLM), which evades both prM- and FL-Abs but retains sensitivity to other type-specific and quaternary cross-reactive (CR) Abs. CR serum from heterotypic (DENV4)-infected non-human primates (NHP) showed lower neutralization titers against D2-FL and D2-FLM than isogenic wildtype DENV2 while similar neutralization titers were observed in serum from homotypic (DENV2)-infected NHP. We propose D2-FL and D2-FLM as valuable tools to delineate CR Ab subtypes in serum as well as an exciting platform for safer live-attenuated DENV vaccines suitable for naïve individuals and children.

## eLife assessment

This **valuable** study describes engineered dengue virus variants that can be used to dissect epitope specificities in polyclonal sera, and to design candidate vaccine antigens that dampen antibody responses against undesirable epitopes. While the major claims are supported by **solid** evidence, experiments to distinguish the impact on antibody binding from neutralizing activities would have strengthened the study. This work will be of interest to virologists and structural biologists working on antibody responses to flaviviruses.

## Introduction

Dengue virus (DENV) is a member of the *Flavivirus* genus and is a major global public health threat, with four major serotypes of DENV found worldwide. Dengue causes ~400 million infections each year, of which ~20% of cases present clinically, a subset of which may progress to severe dengue hemorrhagic fever/dengue shock syndrome (DHF/DSS) (**Bhatt et al., 2013**; **Brady et al., 2012**). DENV is transmitted through *Aedes* mosquito vectors, and globalization and global warming are increasing the endemic range of dengue worldwide (**Ebi and Nealon, 2016**; **Messina et al., 2019**). The pathogenesis of dengue is complex as first-time infections are rarely severe and lead to serotype-specific

*For correspondence:
victor.tse@health.slu.edu

immunity. However, reinfection with a different serotype increases the risk of developing DHF/DSS (*Halstead, 1988*). This is thought to be due to the phenomenon of antibody-dependent enhancement (ADE), in which poorly neutralizing cross-reactive (CR) antibodies (Abs) lead to enhanced viral uptake and infection of unique cell populations in an Fcγ-receptor-mediated manner (*Katzelnick et al., 2017*).

ADE remains a major challenge for DENV vaccine development (*Rey et al., 2018*). The leading DENV vaccine platforms in clinical testing are tetravalent live-attenuated virus mixtures of all four serotypes. However, creating formulations that elicit a balanced response has proven challenging (*Rabaa et al., 2017*). Additionally, lab-grown strains differ from patient-derived DENVs in both maturation status and antigenicity (*Raut et al., 2019*). In particular, Abs targeting the fusion loop (FL) have been reported to neutralize lab and patient strains with differing strengths and have been observed to facilitate Fcγ-receptor uptake in vitro and therefore ADE (*Lai et al., 2013*; *Beltramello et al., 2010*; *Costin et al., 2013*). Currently, there is a single FDA-approved DENV vaccine, Dengvaxia. However, it is only approved for use in individuals aged 9–16 with previous DENV infection living in endemic areas and is contraindicated for use in naïve individuals and younger children. In naïve children, vaccination stimulated non-neutralizing CR Abs that increased the risk of severe disease after DENV infection (*Wilder-Smith, 2019*; *Anonymous, 2019*). Other DENV vaccines have been tested or are currently undergoing clinical trial, but thus far none have been approved for use in the United States (*Izmirly et al., 2020*). The vaccine Qdenga has been approved in the European Union, Indonesia, and Brazil, although vaccine efficacy in adults, naïve individuals, and with all serotypes has not yet been shown (*Angelin et al., 2023*).

The DENV FL is located in Envelope (E) protein domain II (EDII) and is involved in monomer–monomer contacts with EDIII (*Modis et al., 2004*). During the DENV infection cycle, low pH triggers a conformational change in the E protein (*Klein et al., 2013*). The structure of the virion rearranges, and individual monomers form a trimer with all three FLs in the same orientation, ready to initiate membrane fusion (*Modis et al., 2004*; *Klein et al., 2013*). The core FL motif (DRGWGNGCGLFGK, AA 98–110) is highly conserved, with 100% amino acid conservation in all DENV serotypes and other mosquito-borne flaviviruses, including yellow fever virus (YFV), Zika virus (ZIKV), West Nile virus (WNV), Kunjin virus (KUNV), Murray Valley encephalitis virus (MVEV), Japanese encephalitis virus (JEV), Usutu virus (USUV), and Saint Louis encephalitis virus (SLEV; *Figure 1A*). Although the extreme conservation and critical role in entry have led to it being considered difficult to alter the antigenic epitope by changing more than one amino acid of the FL, we successfully tested the hypothesis that massively parallel-directed evolution could produce viable DENV FL mutants that were still capable of fusion and entry, while altering the antigenic footprint. The FL mutations, in combination with optimized prM cleavage site mutations, ablate neutralization by the prM- and FL-Abs, retain sensitivity to other protective Abs, and provide a novel vaccine strategy for DENV.

## Results

To engineer a virus with a novel antigenic footprint at the FL, we targeted the core conserved FL motif. We generated two different saturation mutagenesis libraries, each with five randomized amino acids: DRG**X**G**X**G**XXX**FGK (Library 1; AA 101, 103, 105–107) and DRG**XXXXX**GLFGK (Library 2 AA 101–105). Library 1 was designed to mutate known residues targeted by FL mAbs while Library 2 focused on a continuous linear peptide that is the epitope for FL-Abs to maximally alter antigenicity (*Smith et al., 2013*). Saturation mutagenesis plasmid libraries were used to produce viral libraries in either C6/36 (*Aedes albopictus* mosquito) or Vero 81 (African green monkey) cells and passaged three times in their respective cell types. Following directed evolution, viral genomes were extracted and subjected to deep sequencing to identify surviving and enriched variants (*Figure 1B*). Due to the high level of conservation, it was not surprising that most mutational combinations failed to yield viable progeny. In fact, evolutions carried out on Library 2 only yielded wildtype sequences. For Library 1, wildtype sequences dominated in Vero 81-evolved libraries. However, a novel variant emerged in C6/36 cells with two amino acid changes: DRGWG**S**GC**L**LFGK. The major variant comprised ~95% of the population, while the next most populous variant (DRGWG**S**GC**W**LFGK) comprised only 0.25% (*Figure 1C*). Bulk Sanger sequencing revealed an additional Env-T171A mutation outside of the FL region. Residues W101, C105, and L107 were preserved in our final sequence, supporting the importance of these residues (*Modis et al., 2004*). When modeled on the pre-fusion DENV2 structure, the N103S and G106L mutations are located at the interface with the neighboring monomer EDIII

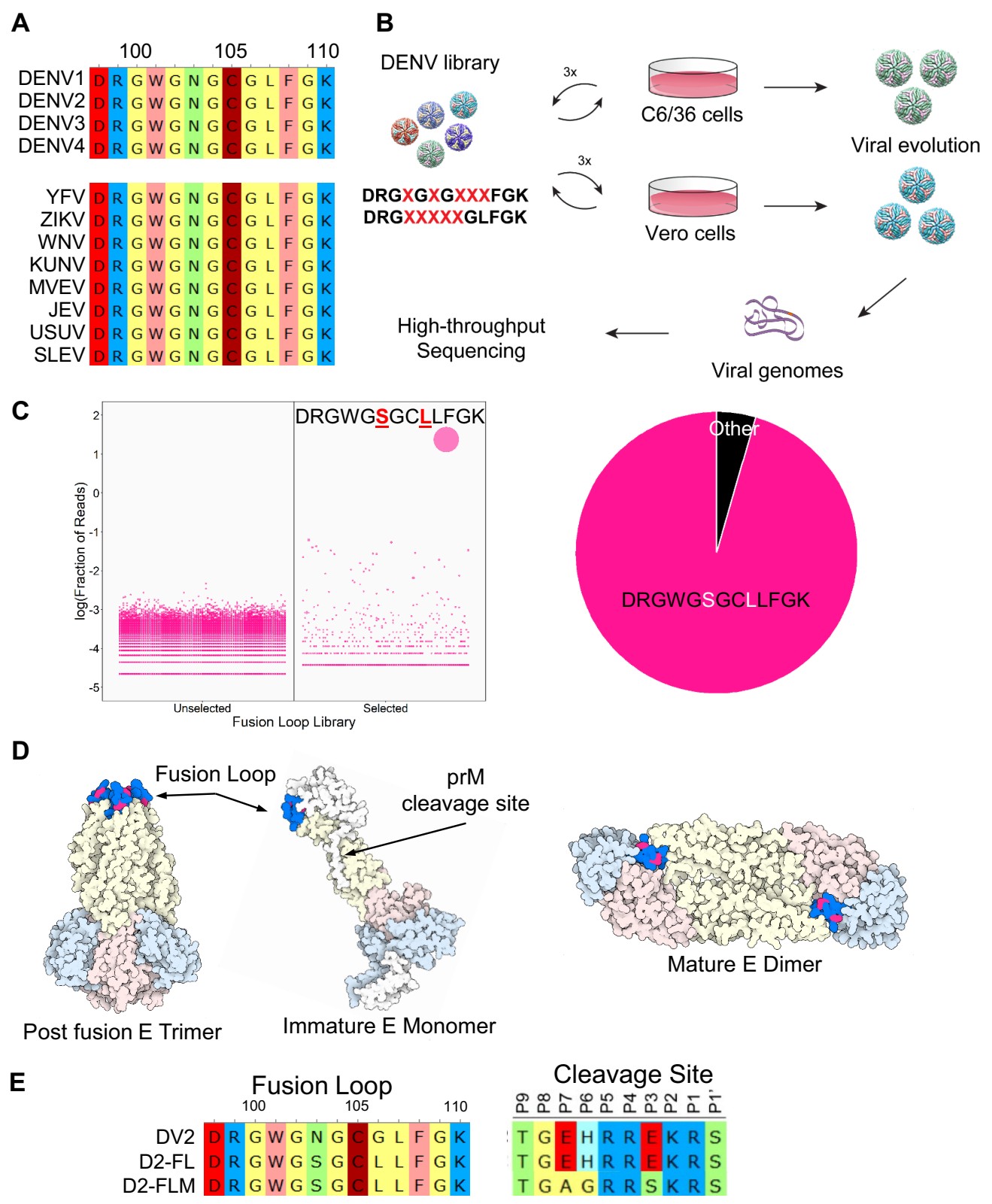

**Figure 1.** Generation of DENV2 fusion loop mutants via directed evolution. (**A**) Alignment of top: dengue virus fusion loops; bottom: mosquito-borne flavivirus fusion loops, including yellow fever virus (YFV), Zika virus (ZIKV), West Nile virus (WNV), Kunjin virus (KUNV), Murray Valley encephalitis virus (MVEV), Japanese encephalitis virus (JEV), Usutu virus (USUV), and Saint Louis encephalitis virus (SLEV). Amino acids are colored by functional groups: negatively charged (red), positively charged (blue), nonpolar (yellow), polar (green), aromatic (pink), and sulfide (dark red). (**B**) Schematic of directed

Figure 1 continued

evolution procedure. Saturation mutagenesis libraries were used to produce viral libraries, which were passaged three times in either C6/36 or Vero 81 cells. At the end of the selection, viral genomes were isolated and mutations were identified by high-throughput sequencing. (C) Left: bubble plot of the sequences identified from either the unselected or selected (passage 3) C6/36 DENV libraries. Right: pie chart of the sequences from passage 3 C6/36 DENV libraries. (D) Structure of the DENV envelope with the fusion loop mutations highlighted in red. (E) Sequences of the fusion loop and furin cleavage site of DENV2, D2-FL, and D2-FLM.

domain, protected from the aqueous environment. In the post-fusion form, the two residues are located between W101 and F108 and form the bowl concavity above the chlorine ion in the post-fusion trimer (*Figure 1D*). We used reverse genetics to re-derive the FL N103S/G106L/T171A mutant, which we term D2-FL. As enhancing Abs also target prM (*Rodenhuis-Zybert et al., 2010*), we also created a mature version of D2-FL termed D2-FLM, containing both the evolved FL motif and our previously published evolved prM furin cleavage site, which results in a more mature virion like those found in infected patients (*Figure 1E*; *Raut et al., 2019*; *Tse et al., 2022*).

We performed growth curves comparing DENV2, D2-FL, and D2-FLM in both C6/36 and Vero 81 cells. In C6/36 cells, the growth of all three viruses was comparable, reaching high titers of $10^6$–$10^7$ FFU/mL. However, in Vero 81 cells, both FL mutant viruses were highly attenuated, with a 2–2.5 log reduction in titer (*Figure 2A*). The species-specific phenotype in culture involved a change from insect to mammalian cells, as well as a change in growth temperature. To investigate whether the mutant viruses were more unstable at higher temperatures, we performed a thermostability assay, comparing viruses incubated at temperatures ranging from 4 to 55°C before infection. The three viruses had comparable thermostabilities, indicating that this does not explain the attenuation of the FL mutants (*Figure 2B*). Because the D2-FLM virus contains mutations that increase prM cleavage frequency, we also assayed the maturation status of the three viruses by western blot. D2-FL had a comparable prM:E ratio to the isogenic wildtype DENV2 (DV2-WT), while, as expected, D2-FLM had a reduced prM:E ratio, indicating a higher degree of maturation (*Figure 2C*).

Next, we characterized the ability of Abs targeting the FL to recognize DV2-WT, D2-FL, and D2-FLM with a panel of monoclonal antibodies (mAbs). Importantly, D2-FL and D2-FLM were resistant to mAbs targeting the FL. Neutralization by 1M7 is reduced by ~2 logs in both variants, 1N5 neutralization is reduced by ~1 log for D2-FL and reduced to background levels for D2-FLM, and no neutralization was observed for 1L6 or 4G2 for either variant (*Figure 3A*; *Smith et al., 2013*). Focusing on the D2-FLM virus containing both evolved motifs, we then characterized the antigenicity of the whole virion with a panel of mAbs. As expected, D2-FLM was unable to be neutralized by the prM Abs 1E16 and 5M22; the Ab 2H2 does not neutralize either DV2-WT or D2-FLM (*Figure 3B*). For Abs targeting epitopes in non-mutated regions, including the ED1 and EDE epitopes that target EDII and EDIII, $FRNT_{50}$ values were generally comparable, although EDE1-C10 shows a moderate but statistically significant reduction between DV2-WT and D2-FLM, indicating that the overall virion structural integrity was intact (*Figure 3B*).

Next, we analyzed neutralization of the D2-FLM virus using serum derived from convalescent humans and experimentally infected rhesus macaque non-human primates (NHPs). Overall, we tested serum from three different cohorts with a total of 6 humans and 9 NHPs at different time points with a total of 27 samples. In the first cohort (NHP infected with $1 \times 10^6$ FFU Vero-grown virus via subcutaneous [SC] injection) (*Young et al., 2023*), serum from homotypic DENV2-infected NHPs (n = 3) did not display a difference in neutralization between DV2-WT, D2-FL, and D2-FLM, confirming that prM and FL epitopes are not significant contributors to the homotypic type-specific (TS) neutralizing Ab response in primates (*Figure 3C*). In two NHPs infected with DENV4, strong neutralization potency ($FRNT_{50}$ between 1:100 and 1:1000) was demonstrated against DV2-WT (*Figure 3C*). Heterologous cross-neutralization was significantly reduced to background levels ($FRNT_{50}$ < 1:40) against the D2-FLM virus at 90 days post-infection (dpi). Neutralization observed against D2-FL, in general, fell between DV2-WT and D2-FLM. NHP 0Y0 displayed little difference in neutralization between D2-FL and D2-FLM, suggesting that FL Abs were more prominent in this animal. Interestingly, in animal 3Z6, at 20 dpi, neutralization against DENV-FL was comparable to DV2-WT, while D2-FLM was greatly reduced, indicating that Abs in the sera targeting the immature virion formed a large portion of the CR response. These data suggest that after a single infection many of the CR Ab responses

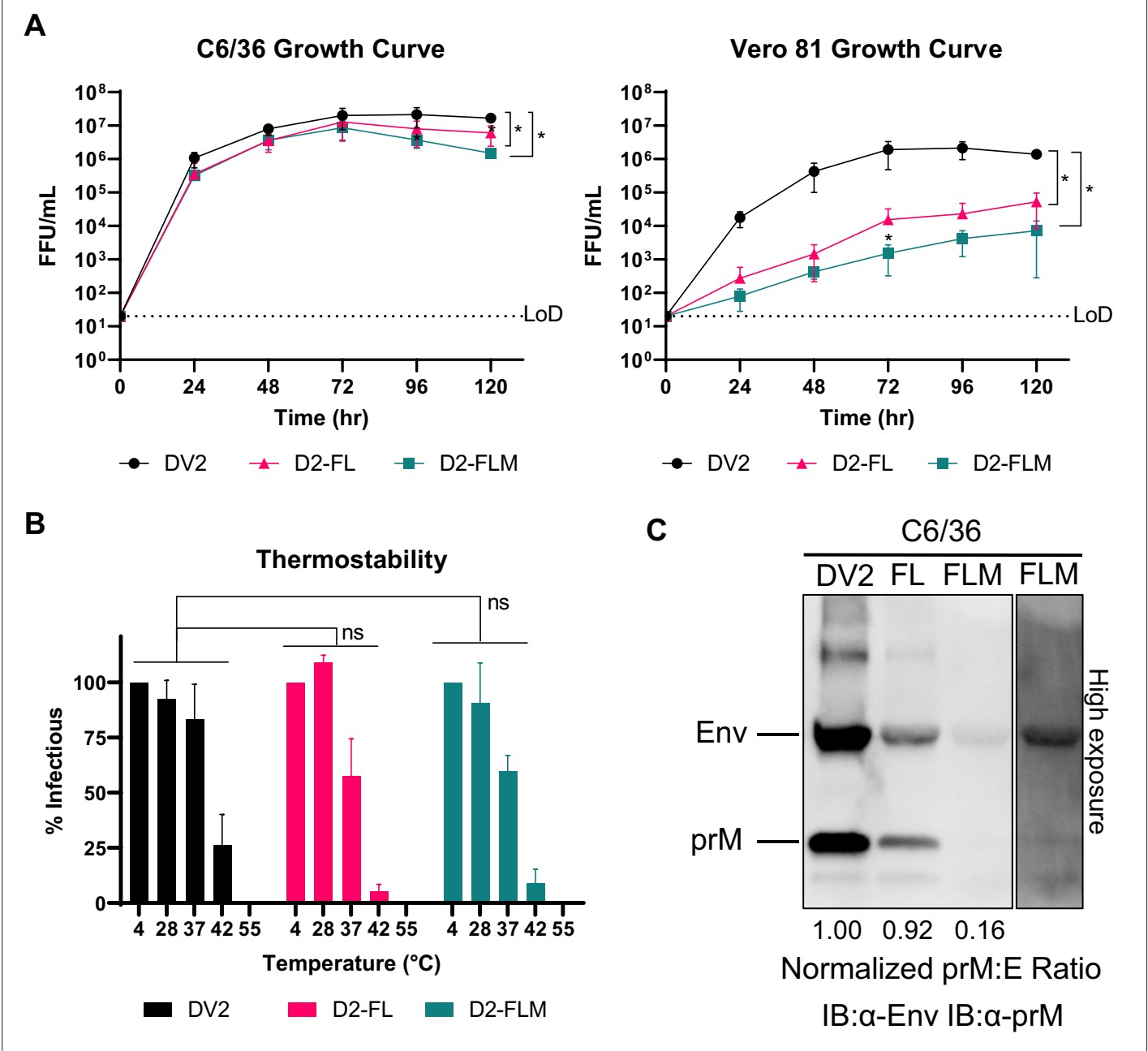

**Figure 2.** Biological and physical properties of mature DENV2 fusion loop mutants. (**A**) Multistep growth curves [Multiplicity of infection (MOI) = 0.05–0.1] of DV2-WT, D2-FL, and D2-FLM on C6/36 cells (left) or Vero 81 cells (right). (**B**) Thermostability assay on DV2-WT, D2-FL, and D2-FLM. (**C**) Western blot of virions, blotted against Envelope and prM proteins. Equal volumes of supernatants from viral infected C6/36 cells at 5 days post-infection (dpi) (**A**) were loaded into each lane. The prM:E ratio was determined and normalized to the DV2-WT ratio. High exposure of D2-FLM (right) illustrates the relative abundance of prM protein to E protein. Averages of three biological replicates are shown. The data are graphed as means ± standard deviations (Error bars). Two-way ANOVA was used for statistical comparison of growth curves and thermostability: ns, not significant; *<0.05; **<0.005; ***<0.0005.

The online version of this article includes the following source data for figure 2:

**Source data 1.** Raw gel images of the western blot shown in *Figure 2C*.

target prM and the FL and the reliance on these Abs for heterotypic neutralization increases over time (*Figure 3C*).

In the second cohort (NHPs infected with 5 × 10⁵ FFU C6/36-grown virus via SC injection) and the third cohort (human serum tested positive by ELISA for homotypic DENV infection) (*Lopez et al., 2021*), most of the serum (18/24) did not cross-neutralize DV2-WT (FRNT$_{50}$ < 1:40), leading

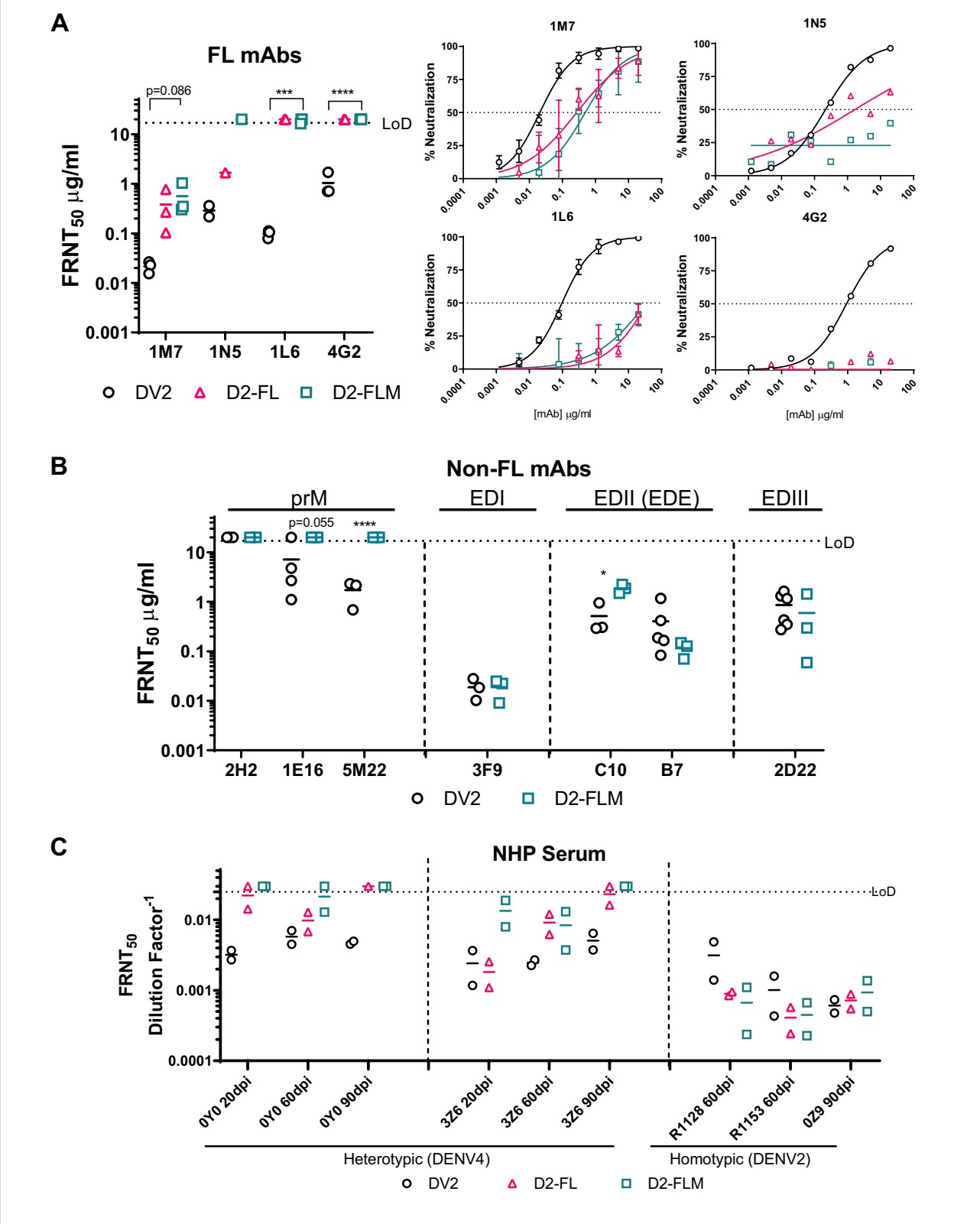

**Figure 3.** Fusion loop (FL) mutant is insensitive to FL monoclonal antibodies (mAbs), the major target for cross-reactive Abs in non-human primates (NHPs). (**A**) Left: FRNT$_{50}$ values for neutralization of DV2-WT, D2-FL, and D2-FLM with mAbs against the FL (1M7, 1N5, 1L6, 4G2). All Abs were tested in at least n = 3 independent experiments, except 1N5 due to limited Ab. Right: average neutralization curves for neutralization of DV2-WT, D2-FL, and D2-FLM with mAbs against the DENV2 FL. (**B**) FRNT$_{50}$ values for neutralization of DV2-WT and D2-FLM with mAbs against DENV2 prM (2H2, 1E16,

*Figure 3 continued on next page*

*Figure 3 continued*

5M22), EDI (3F9), EDE (C10, B7), and EDIII (2D22). All Abs were tested in at least n = 3 independent experiments. The data are graphed as means ± standard deviations (Error bars). (**C**) Neutralization of DV2-WT, D2-FL, and D2-FLM with sera from NHPs infected with either DENV4 or DENV2. $FRNT_{50}$s were compared using Student's *t*-test. Significant symbols are as follows: *p<0.05; **p<0.005; ***p<0.0005; ****p<0.00005. The data are graphed as means ± standard deviations.

to inconclusive result in determining the CR-Ab population in these sera (*Table 1*). The difference between the three cohorts also highlights the heterogeneity of Ab composition in humans and NHPs after DENV exposure. Nevertheless, the collection of FL, mature, and FL-mature variants provides new opportunity to delineate antibody composition in complex polyclonal serum from DENV natural infection and vaccination.

## Discussion

Mechanistic understanding of vaccine protection and identification of correlates of protection are immensely important for DENV vaccine development. The dual protective and enhancing properties of DENV Abs create major challenges for dissecting the role of various Ab populations in disease protection. Cross-reactive weakly neutralizing prM- and FL-Abs are often immunodominant after primary DENV infection (*Lai et al., 2013*; *Smith et al., 2012*; *Oliphant et al., 2006*; *Lai et al., 2008*; *Dejnirattisai et al., 2010*), and can lead to overestimation of the levels of heterotypic protection in traditional neutralization assays. Since these same antibodies are also associated with ADE (*Rodenhuis-Zybert et al., 2010*), inaccurate conclusions could have dire consequences if protection in vitro translates to the enhancement of disease in human vaccinees. Unfortunately, Ab profiling in polyclonal serum is mainly performed by ELISA, and a neutralization assay that can discriminate Abs does not exist. The D2-FLM variant is not neutralized by FL- and prM-mAbs and appears insensitive to neutralization by these Abs in polyclonal serum. Of note, EDE1-C10 neutralization potency was also reduced in our FL-M variant, further indicating our mutations are affecting the tip of the EDII region that partially overlaps with the EDE1 epitope (*Sharma et al., 2021*). In combination with other chimeric DENVs (*Andrade et al., 2019*; *Young et al., 2020*; *Gallichotte et al., 2015*), D2-FLM provides a reagent to distinguish between TS, protective CR (e.g. E dimer epitope [EDE]) (*Barba-Spaeth et al., 2016*; *Dejnirattisai et al., 2015*) and ADE-prone CR (e.g. FL and prM) (*Dejnirattisai et al., 2010*) Ab subclasses in neutralization assays after infection and vaccination.

Due to the ADE properties of DENV Abs, studies to understand and eliminate ADE phenotype are under active investigation. For example, mAbs can be engineered to eliminate binding to the Fcγ receptor, abolishing ADE (*Kotaki et al., 2021*). While this methodology holds potential for Ab therapeutic development and passive immunization strategies, it is not relevant for vaccination. As FL and prM targeting Abs are the major species demonstrated to cause ADE in vitro, we and others hypothesized that these Abs are responsible for ADE-driven negative outcomes after primary infection and vaccination (*Lai et al., 2013*; *Beltramello et al., 2010*; *Costin et al., 2013*; *de Alwis et al., 2014*). We propose that genetic ablation of the FL and prM epitopes in vaccine strains will minimize the production of these subclasses of Abs responsible for undesirable vaccine responses. Indeed, covalently locked E-dimers and E-dimers with FL mutations have been engineered as potential subunit vaccines that reduce the availability of the FL, thereby reducing the production of FL Abs (*Rouvinski et al., 2017*; *Rockstroh et al., 2015*; *Slon-Campos et al., 2019*; *Metz et al., 2017*). DENV subunit vaccines are an area of active study (*Kudlacek et al., 2021*); however, monomer/dimer subunits can also expose additional, interior-facing epitopes not normally exposed to the cell. Furthermore, dimer subunits are not a complete representation of the DENV virion, which presents other structurally important interfaces such as the threefold and fivefold symmetries. Concerns about balanced immunity to all four serotypes also apply to subunit vaccine platforms. Given the complexity of the immune response to DENV, live virus vaccine platforms have thus far been more successful. However, the FL is strongly mutationally intolerant. Previous reported mutations in the FL are mainly derived from experimental evolution using FL-Abs to select for escape mutant or by deep mutational scanning (DMS) of the Env protein for Ab epitope mapping. Mutations in the FL epitope were observed in a DENV2-NGC-V2 (G106V) (*Goncalvez et al., 2004*), attenuated JEV vaccine strain SA14-14-2 (L107F) (*Gromowski et al., 2015*), and attenuated WNV-NY99 (L107F) (*Zhang et al., 2006*). While most of

the mutations, including the double mutations reported here lead to attenuation of the virus, a recent DMS study showed that Zika-G106A has no observable impact on viral fitness (*Chambers et al., 2018*). Interestingly, we also recovered a mutation G106L, suggesting that positions 103, 106, and 107 are tolerable FL positions for mutation in mosquito-borne flavivirus. On the other hand, tick-borne flaviviruses, no known vector flaviviruses, and invertebrate-specific flaviviruses show a more diverse FL composition. The inflexibility of mosquito-borne flaviviruses might be due to the evolutionary constraint of the virus to switch between mosquito and vertebrate hosts. Using directed evolution, we successfully generated our D2-FLM variant that combines viability with the desired Ab responses. Therefore, the D2-FLM variant is a novel candidate for a vaccine strain, which presents all the native structures and complex symmetries of DENV necessary for T-cell-mediated responses and which can elicit more optimal protective Ab responses (*Tian et al., 2019*; *Elong Ngono and Shresta, 2019*; *Waickman et al., 2019*).

Other considerations of high importance when designing a live DENV vaccine include strain selection and serotype balance (*Gallichotte et al., 2018*). In the current study, we used DENV2 S16803, a prototype for DENV2 (*Halstead and Marchette, 2003*). However, S16803 was isolated several decades ago, and it may be beneficial to utilize more contemporaneous strains (*Rabaa et al., 2017*; *Juraska et al., 2018*). Work is currently ongoing to demonstrate the portability of the evolved FL motif on additional DENV2 strains and other serotypes, which is essential for tetravalent vaccine production. D2-FLM was highly attenuated in Vero cells, creating a challenge for vaccine production. Therefore, further adaptation of this strain to grow efficiently in mammalian cells while retaining its antigenic properties is needed. We have tested a panel of FL-Abs; however, we cannot exclude the possibility that other FL-Abs may not be affected by N103S and G106L. Our study confirmed that saturation mutagenesis could generate mutants with multiple amino acid changes, and we are currently using D2-FLM as backbone to iteratively evolve additional mutations in FL to further vary the FL antigenic epitope. Taken together, the FLM variant holds new possibilities for a new generation of DENV vaccines, as well as a platform to readily measure TS and CR ADE-type responses and thereby assess the true protective potential of DENV vaccine trials and safeguard approval of DENV vaccines for human use.

## Materials and methods
### Cells and viruses
*A. albopictus* C6/36 (ATCC CRL-1660) cells were grown in MEM (Gibco) with 5% FBS (HyClone), 1% penicillin/streptomycin (Gibco), 0.1 mM nonessential amino acids (Gibco), 1% HEPES (Gibco), and 2 mM GlutaMAX (Gibco), cultured at 32°C with 5% $CO_2$. African green monkey Vero 81 cells (ATCC CCL-81) were grown in DMEM/F12 (Gibco) with 10% FBS, 1% penicillin/streptomycin (Gibco), 0.1 mM nonessential amino acids (Gibco), and 1% HEPES (Gibco), cultured at 37°C with 5% $CO_2$. Cells were tested negative for mycoplasma. DENV viruses were grown in C6/36 or Vero 81 cells maintained in infection media. C6/36 infection media consists of Opti-MEM (Gibco) with 2% FBS (HyClone), 1% penicillin/streptomycin (Gibco), 0.1 mM nonessential amino acids (Gibco), 1% HEPES (Gibco), and 2 mM GlutaMAX (Gibco). Vero 81 infection media consists of DMEM/F12 (Gibco) with 2% FBS, 1% penicillin/streptomycin (Gibco), 0.1 mM nonessential amino acids (Gibco), and 1% HEPES (Gibco). DENV2 strain S16803 was used in this study (*Halstead and Marchette, 2003*). Sequences used for the alignments include DENV1 WestPac-74 (U88535.1), DENV2 S-16803 (GU289914.1), DENV3 3001 (JQ411814.1), DENV4 Sri Lanka-92 (KJ160504.1), YFV 17D (NC_002031.1), SLEV Kern217 (NC_007580.2), JEV (NC_001437.1), USUV Vienna-2001 (NC_006551.1), MVEV (NC_000943.1), and WNV-1 NY99 (NC_009942.1), and ZIKV MR-766 (NC_012532.1).

### DENV reverse genetics
DENV2 S16803 was used in this study. Recombinant viruses were created using a four-plasmid system as previously described (*Messer et al., 2012*), consisting of the DENV genome split into four segments, each cloned into a separate plasmid. The DENV plasmids were digested and ligated to form a single template for in vitro transcription. The resulting RNA was electroporated into either C6/36 or Vero cells. Virus-containing supernatant was harvested at 4–5 d post electroporation and passaged. DENV variants were created through site-directed mutagenesis of the DENV plasmids.

**Table 1.** Summary of FRNT$_{50}$ values of human convalescent serum and non-human primates (NHP) infection serum against DV2, D2-FL (not determined in human serum) and D2-FLM.
FRNT$_{50}$ values were derived from an eight-point dilution curve of serum started from 1:40 with subsequent threefold dilutions. FRNT$_{50}$ values were calculated/extrapolated from the neutralization curves. For human serum, the average FRNT$_{50}$ values were shown from two independent experiments with technical replicates. For NHP serum, FRTN$_{50}$ values were average of three independent experiments. ND, not determined. * denotes the Hill slopes of all the neutralization curve of them sample are <0.5 (low confidence of neutralization).

| Human # | Infected serotype | Collection tme | DV2 | D2-FL | D2-FLM |
|---|---|---|---|---|---|
| DS1500 | DENV1 | Convalescent serum | 1:75 | ND | 1:53 |
| DS2499 | | | 1:80 | ND | 1:52 |
| DS1136 | DENV3 | | 1:133 | ND | 1:45 |
| DS1160 | | | 1:86 | ND | 1:208 |
| DS0275 | DENV4 | | 1:94 | ND | 1:94 |
| DS2239 | | | 1:156 | ND | 1:42 |

| NHP # | Infected serotype | Collection time (dpi) | DV2 | D2-FL | D2-FLM |
|---|---|---|---|---|---|
| R628 | | 30 | <1:40 | <1:40 | <1:40 |
| | | 60 | <1:40 | <1:40 | <1:40 |
| | | 90 | 1:42* | <1:40 | <1:40 |
| R737 | DENV4 | 30 | <1:40 | <1:40 | <1:40 |
| | | 60 | <1:40 | <1:40 | <1:40 |
| | | 90 | <1:40 | <1:40 | <1:40 |
| R1132 | | 30 | <1:40 | <1:40 | <1:40 |
| | | 60 | <1:40 | <1:40 | <1:40 |
| | | 90 | <1:40 | <1:40 | <1:40 |
| R1160 | | 30 | 1:58* | <1:40 | <1:40 |
| | | 60 | 1:57* | 1 : 50* | <1:40 |
| | | 90 | <1:40 | 1 : 57* | 1:91* |

## Library generation and directed evolution

DENV FL libraries were generated through saturation mutagenesis of the indicated resides, based on a previously published protocol (*Tse et al., 2022*; *Tse et al., 2017*). Degenerate NNK oligonucleotides were used to amplify the region, generating a library of mutated DNA fragments. Q5 DNA Polymerase was used with less than 18 cycles to maintain accuracy. The resulting library was cloned into the DENV reverse genetics system. The ligated plasmids were electroporated into DH10B ElectroMax cells (Invitrogen) and directly plated on 5,245 mm$^2$ dishes (Corning) to avoid bias from suspension culture. Colonies were pooled and purified using a Maxiprep kit (QIAGEN), and the plasmid library used for DENV reverse genetics (above). Viral libraries were passaged three times in the corresponding cell type.

## High-throughput sequencing and analysis

Viral RNA was isolated with a QIAamp viral RNA kit (QIAGEN), and cDNA produced using the Superscript IV Reverse Transcriptase (Invitrogen). Amplicons were prepared for sequencing using the Illumina TruSeq system with two rounds of PCR using Q5 Hot Start DNA polymerase (NEB). For the first round of PCR, primers were specific to the DENV2 E sequence surrounding the FL motif with overhangs for the Illumina adapters (RT: CGCAGCTAGAATCGATCTAGCNNNNNNNNNNNNNNNNNNTGTG CACCAGCCAAGC; F:CCCTACACGACGCTCTTCCGATCTNNNNNCAAACCAACATTGGATTTTG

AACTG; R:GACTGGAGTTCAGACGTGTGCTCTTCCGATCTNNNNN<u>CGCAGCTAGAATCGATCTAGC</u>). After purification, this product was used as the template for the second round of PCR using Illumina P5 and P7 primers containing 8-nucleotide indexes (P5: CAAGCAGAAGACGGCATACGAGAT NNNNNNNNGTGACTGGAGTTCAGAC; P7: AATGATACGGCGACCACCGAGATCTACACTCTTTCC CTACACGACGCTCTTCCG). Purified PCR products were analyzed on a Bioanalyzer (Agilent Technologies) and quantified on a Qubit 4 fluorometer (Invitrogen). Amplicon libraries were run on a MiSeq system with 2 × 150 bp reads. Plasmid and P0 libraries were sequenced at a depth of ~4.5 million reads; later passages were sequenced at a depth of ~750,000 reads. Custom Perl and R scripts were used to analyze and plot the data as previously published (*Tse et al., 2022*) and can be found the Tse Lab GitHub site (*Meganck, 2022*).

## DENV growth kinetics

One day before infection, $5 \times 10^5$ cells were seeded in every well of a 6-well plate. Cells with infected with an MOI of 0.05–0.1, estimating $1 \times 10^6$ cells on the day of infection. Infection was carried out for 1 hr in the incubator, followed by 3× washes with PBS and replenishment with fresh infection medium. Then, 300 uL of viral supernatant was collected at 0, 24, 48, 72, 96, and 120 hr and stored at –80°C. All experiments were performed independently at least three times (biological replicate).

## DENV focus-forming assay

Titers of viral supernatant were determined using a standard DENV focus-forming assay. In brief, cells were seeded at $2 \times 10^4$ cells per well of a 96-well plate 1 d before infection. The next day, 50 uL of tenfold serial dilution of viral supernatant were added to each well for 1 hr in the incubator. Afterward, 125 uL of overlay (Opti-MEM, 2% FBS, 1% NEAA, 1% P/S, and 1% methylcellulose) was added to each well. Infection was allowed to continue for 48 hr in the incubator. Overlay was removed, and each well rinsed 3× with PBS followed by a 30 min fixation with 10% formalin in PBS. Cells were blocked in permeabilization buffer (eBioscience) with 5% nonfat dried milk. Primary Abs anti-prM 2H2 and anti-E 4G2 from nonpurified hybridoma supernatant were used at a 1:500 dilution in blocking buffer. Goat anti-mouse HRP secondary (SeraCare KPL 54500011) were used at a 1:1000 dilution in blocking buffer. Followed washing, foci were developed using TrueBlue HRP substrate (SeraCare) and counted using an automated Immunospot analyzer (Cellular Technology).

## Thermal stability assay

The indicated viruses were thawed and incubated at temperatures ranging from 4 to 55°C for 1 hr. Viral titers were determined by focus-forming assay as described above. Viral titers were normalized to sample incubated at 4°C to calculate % infection. All experiments were performed independently at least three times (biological replicate).

## Western blotting

Viral supernatants were combined with 4× Laemmli Sample Buffer (Bio-Rad) and boiled at 95°C for 5 min. After SDS-PAGE electrophoresis, samples were transferred to PVDF membrane and blocked in 3% nonfat milk in PBS-T. A polyclonal rabbit anti-prM (1:1000; Invitrogen PA5-34966) and polyclonal rabbit anti-Env (Invitrogen PA5-32246) in 2% BSA in PBS-T were incubated on the blot for 1 hr at 37°C. Goat anti-rabbit HRP (1:10,000 Jackson ImmunoLab 111-035-144) in 3% milk in PBS-T was incubated on the blot for 1 hr at room temperature. Blots were developed by SuperSignal West Pico Plus chemiluminescent substrate (Thermo Fisher). Blots were imaged on an iBright FL1500 imaging system (Invitrogen). The pixel intensity of individual bands was measured using ImageJ, and the relative maturation was calculated by using the following equation: (prMExp/EnvExp)/(prMWT/EnvWT). All experiments were performed independently at least three times (biological replicate).

## FRNT assay

Focus reduction neuralization titer (FRNT) assays were performed as described previously with C6/36 cells (*Tse et al., 2022*). $1 \times 10^5$ cells were seeded in a 96-well plate the day prior to infection. Abs or sera were serially diluted and mixed with virus (~100 FFU/well) at a 1:1 volume and incubated for 1 hr in the incubator. The mixture was added onto the plate with cells and incubated for 1 hr in the incubator, then overlay was added (see 'Focus-forming assay') and plates were incubated for

48 hr. Viral foci were stained and counted as described above ('Focus-forming assay'). A variable slope sigmoidal dose–response curve was fitted to the data, and values were calculated with top or bottom restraints of 100 and 0 using GraphPad Prism version 9.0. All experiments were performed independently at least two times due to limited amounts of human serum. For serum neutralization, $FRNT_{50}$ values were derived from an eight-point dilution curve started from 1:40 with subsequent threefold dilutions. $FRNT_{50}$ values were calculated/extrapolated from the neutralization curves. For human serum, the average $FRNT_{50}$ values were shown from two independent experiments (biological replicate) with technical replicates. For NHP serum, $FRTN_{50}$ values were the average of three independent experiments (biological replicate).

### Statistical analysis

GraphPad Prism version 9.0 was used for statistical analysis. Titer and % infection of D2-FL and D2-FLM were compared to the DV2 using two-way ANOVA. $FRNT_{50}$s were compared using Student's *t*-test. Significant symbols are as follows: $*p < 0.05$; $**p < 0.005$; $***p < 0.0005$; $****p < 0.00005$. The data are graphed as means ± standard deviations (Error bars).

### Materials availability statement

Materials generated from this study are available upon request.

## Acknowledgements

We thank members of the Tse, Baric, and DeSilva laboratories for helpful discussions. This work was supported by NIAID R01AI107731 to AD and RSB, P01AI106695 to RSB, and NIAID F30AI160898 to DRZ. LJW is supported by NIAID P01 5112869.

## Additional information

### Competing interests

Rita M Meganck: inventor on a patent application (#63/320,922) filed on the subject matter of the manuscript. Ralph S Baric: member of the advisory board of VaxArt and Invivyd and has collaborations with Takeda, Pfizer, Moderna, Ridgeback Biosciences, Gilead, and Eli Lily. Inventor on a patent application (#63/320,922) filed on the subject matter of the manuscript. Longping V Tse: Inventor on a patent application (#63/320,922) filed on the subject matter of the manuscript. The other authors declare that no competing interests exist.

### Funding

| Funder | Grant reference number | Author |
| --- | --- | --- |
| National Institute of Allergy and Infectious Diseases | R01AI107731 | Arivianda M DeSilva |
| National Institute of Allergy and Infectious Diseases | P01AI106695 | Arivianda M DeSilva |

The funders had no role in study design, data collection and interpretation, or the decision to submit the work for publication.

### Author contributions

Rita M Meganck, Conceptualization, Data curation, Software, Formal analysis, Validation, Investigation, Visualization, Methodology, Writing – original draft, Writing – review and editing; Deanna Zhu, Stephanie Dong, Data curation, Formal analysis, Investigation; Lisa J Snoderly-Foster, Yago R Dalben, Data curation; Devina Thiono, Laura J White, Resources; Arivianda M DeSilva, Ralph S Baric, Resources, Supervision, Funding acquisition, Project administration; Longping V Tse, Conceptualization,

Resources, Data curation, Formal analysis, Supervision, Funding acquisition, Validation, Investigation, Visualization, Methodology, Project administration, Writing – review and editing

## Author ORCIDs
Devina Thiono https://orcid.org/0000-0002-8330-0396
Arivianda M DeSilva https://orcid.org/0000-0003-3317-5950
Longping V Tse https://orcid.org/0000-0001-7582-8396

Reviewer #1 (Public Review): https://doi.org/10.7554/eLife.87555.3.sa1
Reviewer #2 (Public Review): https://doi.org/10.7554/eLife.87555.3.sa2
Author Response https://doi.org/10.7554/eLife.87555.3.sa3

## Additional files

### Supplementary files
• MDAR checklist

### Data availability
Sequencing data have been deposited in SRA under accession code PRJNA947985. Code used for analysis can be found on the Tse Lab GitHub site (https://github.com/TseLabVirology/Saturation-Mutagenesis-Pipeline copy archived at *Meganck, 2022*).

The following dataset was generated:

| Author(s) | Year | Dataset title | Dataset URL | Database and Identifier |
|---|---|---|---|---|
| Meganck RM, Zhu D, Dong S, Snoderly-Foster LJ, Dalben YR, Thiono D, White LJ, DeSilva AM, Baric RS, Tse LV | 2023 | Evolution of a Functionally Intact but Antigenically Distinct DENV Fusion Loop | https://www.ncbi.nlm.nih.gov/bioproject/PRJNA947985 | NCBI BioProject, PRJNA947985 |

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
