## [Editor Report · eLife assessment]

This **valuable** study describes engineered dengue virus variants that can be used to dissect epitope specificities in polyclonal sera, and to design candidate vaccine antigens that dampen antibody responses against undesirable epitopes. While the major claims are supported by **solid** evidence, experiments to distinguish the impact on antibody binding from neutralizing activities would have strengthened the study. This work will be of interest to virologists and structural biologists working on antibody responses to flaviviruses.

---

## [Referee Report · Reviewer #1 (Public Review)]

Summary of the major findings -

1. The authors used saturation mutagenesis and directed evolution to mutate the highly conserved fusion loop (98 DRGWGNGCGLFGK 110) of the Envelope (E) glycoprotein of Dengue virus (DENV). They created 2 libraries with parallel mutations at amino acids 101, 103, 105-107, and 101-105 respectively. The in vitro transcribed RNA from the two plasmid libraries was electroporated separately into Vero and C6/36 cells and passaged thrice in each of these cells. They successfully recovered a variant N103S/G106L from Library 1 in C6/36 cells, which represented 95% of the sequence population and contained another mutation in E outside the fusion loop (T171A). Library 2 was unsuccessful in either cell type.

2. The fusion loop mutant virus called D2-FL (N103S/G106L) was created through reverse genetics. Another variant called D2-FLM was also created, which in addition to the fusion loop mutations, also contains a previously published, evolved, and optimized prM-furin cleavage sequence that results in a mature version of the virus (with lower prM content). Both D2-FL and D2-FLM viruses grew comparably to wild type virus in mosquito (C6/36) cells but their infectious titers were 2-2.5 log lower than wildtype virus when grown in mammalian (Vero) cells. These viruses were not compromised in thermostability, and the mechanism for attenuation in Vero cells remains unknown.

4. Next, the authors probed the neutralization of these viruses using a panel of monoclonal antibodies (mAbs) against fusion loop and domain I, II and III of E protein, and against prM protein. As intended, neutralization by fusion loop mAbs was reduced or impaired for both D2-FL and D2-FLM, compared to wild type DENV2. D2-FLM virus was equivalent to wild type with respect to neutralization by domain I, II, and III antibodies tested (except domain II-C10 mAb) suggesting an intact global antigenic landscape of the mutant virion. As expected, D2-FLM was also resistant to neutralization by prM mAbs (D2-FL was not tested in this batch of experiments).

5. Finally, the authors evaluated neutralization in the context of polyclonal serum from convalescent humans (n=6) and experimentally infected non-human primates (n=9) at different time points (27 total samples). Homotypic sera (DENV2) neutralized D2-FL, D2-FLM, and wild type DENV similarly, suggesting that the contribution of fusion loop and prM epitopes is insignificant in a serotype-specific neutralization response. However, heterotypic sera (DENV4) neutralized D2-FL and D2-FLM less potently than wild type DENV2, especially at later time points, demonstrating the contribution of fusion loop- and prM-specific antibodies to heterotypic neutralization.

Impact of the study-

1. The engineered D2-FL and D2-FLM viruses are valuable reagents to probe antibodies targeting the fusion loop and prM in the overall polyclonal response to DENV.

2. Though more work is needed, these viruses can facilitate the design of a new generation of DENV vaccine that does not elicit fusion loop- and prM-specific antibodies, which are often poorly neutralizing and lead to antibody-dependent enhancement effect (ADE).

3. This work can be extended to other members of the flavivirus family.

4. A broader impact of their work is a reminder that conserved amino acids may not always be critical for function and therefore should not be immediately dismissed in substitution/mutagenesis/protein design efforts.

Appraisal of the results -

The data largely support the conclusions, but some improvements and extensions can benefit the work.

1. In Figure 3A, the authors concluded that the engineered dengue virus fusion loop mutant viruses are insensitive to monoclonal antibodies (mAbs) targeting the fusion loop. However, the reduction in neutralization sensitivity varied depending on the mAb tested. The contribution of the optimized prM cleavage site (D2-FLM) to sensitivity to fusion loop mAbs also varied.

a) Are the epitopes known for these mAbs? It would be useful to discuss how the epitope of 1M7 differs from the other mAbs. What are the critical residues?

d) Maybe the D2-FL mutant can be further evolved with selection pressure with fusion loop mAbs 1M7 +/-1N5 and/or other fusion loop mAbs.

2. It would have been useful to include D2-M for comparison (with evolved furin cleavage sequence but no fusion loop mutations).

3. Data for polyclonal serum can be better discussed. Table 1 is not discussed much in the text.

Suggestions for further experiments-

1. It would be interesting to see the phenotype of single mutants N103S and G106L, relative to double mutant N103S/G106L (D2-FL).

2. The fusion capability of these viruses can be gauged using liposome fusion assay under different pH conditions and different lipids.

3. Correlative antibody binding vs neutralization data would be useful.

---

## [Referee Report · Reviewer #2 (Public Review)]

Antibody-dependent enhancement (ADE) of Dengue is largely driven by cross-reactive antibodies that target the DENV fusion loop or pre-membrane protein. Screening polyclonal sera for antibodies that bind to these cross-reactive epitopes could increase the successful implementation of a safe DENV vaccine that does not lead to ADE. However, there are few reliable tools to rapidly assess the polyclonal sera for epitope targets and ADE potential. Here the authors develop a live viral tool to rapidly screen polyclonal sera for binding to fusion loop and pre-membrane epitopes. The authors performed a deep mutational scan for viable viruses with mutations in the fusion loop (FL). The authors identified two mutations functionally tolerable in insect C6/36 cells, but lead to defective replication in mammalian Vero cells. These mutant viruses, D2-FL and D2-FLM, were tested for epitope presentation with a panel of monoclonal antibodies and polyclonal sera. The D2-FL and D2-FLM viruses were not neutralized by FL-specific monoclonal antibodies demonstrating that the FL epitope has been ablated.

Overall the central conclusion that the engineered viruses can predict epitopes targeted by antibodies is supported by the data and the D2-FL and D2-FLM viruses represent a valuable tool to the DENV research community.

---

## [Author Response]

The following is the authors’ response to the original reviews.

**Reviewer #1 (Public Review):**
Summary of the major findings -1. The authors used saturation mutagenesis and directed evolution to mutate the highly conserved fusion loop (98 DRGWGNGCGLFGK 110) of the Envelope (E) glycoprotein of Dengue virus (DENV). They created 2 libraries with parallel mutations at amino acids 101, 103, 105-107, and 101-105 respectively. The in vitro transcribed RNA from the two plasmid libraries was electroporated separately into Vero and C6/36 cells and passaged thrice in each of these cells. They successfully recovered a variant N103S/G106L from Library 1 in C6/36 cells, which represented 95% of the sequence population and contained another mutation in E outside the fusion loop (T171A). Library 2 was unsuccessful in either cell type.1. The fusion loop mutant virus called D2-FL (N103S/G106L) was created through reverse genetics. Another variant called D2-FLM was also created, which in addition to the fusion loop mutations, also contains a previously published, evolved, and optimized prM-furin cleavage sequence that results in a mature version of the virus (with lower prM content). Both D2-FL and D2-FLM viruses grew comparably to wild type virus in mosquito (C6/36) cells but their infectious titers were 2-2.5 log lower than wild type virus when grown in mammalian (Vero) cells. These viruses were not compromised in thermostability, and the mechanism for attenuation in Vero cells remains unknown.1. Next, the authors probed the neutralization of these viruses using a panel of monoclonal antibodies (mAbs) against fusion loop and domain I, II and III of E protein, and against prM protein. As intended, neutralization by fusion loop mAbs was reduced or impaired for both D2-FL and D2-FLM, compared to wild type DENV2. D2-FLM virus was equivalent to wild type with respect to neutralization by domain I, II, and III antibodies tested (except domain II-C10 mAb) suggesting an intact global antigenic landscape of the mutant virion. As expected, D2-FLM was also resistant to neutralization by prM mAbs (D2-FL was not tested in this batch of experiments).1. Finally, the authors evaluated neutralization in the context of polyclonal serum from convalescent humans (n=6) and experimentally infected non-human primates (n=9) at different time points (27 total samples). Homotypic sera (DENV2) neutralized D2-FL, D2-FLM, and wild type DENV similarly, suggesting that the contribution of fusion loop and prM epitopes is insignificant in a serotype-specific neutralization response. However, heterotypic sera (DENV4) neutralized D2-FL and D2-FLM less potently than wild type DENV2, especially at later time points, demonstrating the contribution of fusion loop- and prM-specific antibodies to heterotypic neutralization.Impact of the study-1. The engineered D2-FL and D2-FLM viruses are valuable reagents to probe antibodies targeting the fusion loop and prM in the overall polyclonal response to DENV.1. Though more work is needed, these viruses can facilitate the design of a new generation of DENV vaccine that does not elicit fusion loop- and prM-specific antibodies, which are often poorly neutralizing and lead to antibody-dependent enhancement effect (ADE).1. This work can be extended to other members of the flavivirus family.1. A broader impact of their work is a reminder that conserved amino acids may not always be critical for function and therefore should not be immediately dismissed in substitution/mutagenesis/protein design efforts.Evaluating this study in the context of prior literature -The authors write "Although the extreme conservation and critical role in entry have led to it being traditionally considered impossible to change the fusion loop, we successfully tested the hypothesis that massively parallel directed evolution could produce viable DENV fusion-loop mutants that were still capable of fusion and entry, while altering the antigenic footprint."".....Previously, a single study on WNV successfully generated a viable virus with a single mutation at the fusion loop, although it severely attenuated neurovirulence. Otherwise, it has not been generated in DENV or other mosquito-borne flaviviruses"The above claims are a bit overstated. In the context of other flaviviruses:A previous study applied a similar saturation mutagenesis approach to the *full length* E protein of Zika virus and found that while the conserved fusion loop was mutationally constrained, some mutations, including at amino acid residue 106 were tolerated (PMID 31511387).The Japanese encephalitis virus (JEV) SA14-14-2 live vaccine strain contains a L107F mutation in the fusion loop (in addition to other changes elsewhere in the genome) relative to the parental JEV SA14 strain (PMID: 25855730).For tickborne encephalitis virus (TBEV-DENV4 chimera), H104G/L107F double mutant has been described (PMID: 8331735)There have also been previous examples of functionally tolerated mutations within the DENV fusion loop:Goncalvez et al., isolated an escape variant of DENV 2 using chimpanzee Fab 1A5, with a mutation in the fusion loop G106V (PMID: 15542644). G106 is also mutated in D2-FL clone (N103S/G106L) described in the current study.In the context of single-round infectious DENV, mutation at site 102 within the fusion loop has been shown to retain infectivity (PMID 31820734).

We thank the reviewer for these comments. We have adjusted the text above to better reflect and credit the prior literature. Text is modified as follows in the discussion session.

“Previous reported mutations in the fusion loop are mainly derived from experimental evolution using FL-Ab to select for escape mutant or by deep mutational scanning (DMS) of the Env protein for Ab epitope mapping. Mutations in the FL epitope were observed in a DENV2-NGC-V2 (G106V)39, attenuated JEV vaccine strain SA14-14-2 (L107F)40, attenuated WNV-NY99 (L107F)41. While most of the mutations, including the double mutations reported here lead to attenuation of the virus. A recent DMS study showed that Zika-G106A has no observable impact on viral fitness42. Interestingly, we also recovered a mutation G106L, suggesting position 106 and 107 might be the most tolerable position for mutation in mosquito borne flavivirus FL. On the other hand, tick borne flavivirus as well as vector only flavivirus show a more diverse FL composition. The inflexibility of mosquito borne flavivirus might be due to the evolution constraint of the virus to switch between mosquito and vertebrate hosts.”

Appraisal of the results -The data largely support the conclusions, but some improvements and extensions can benefit the work.1. Line 92-93: "This major variant comprised ~95% of the population, while the next most populous variant comprised only 0.25% (Figure 1C)".What is the sequence of the next most abundant variant?

The sequence of the next most abundant variant has been added to the text.

1. Lines 94-95: "Residues W101, C105, and L107 were preserved in our final sequence, supporting the structural importance of these residues."L107F is viable in other flaviviruses.

We acknowledge that the L107F mutation has been described in other flaviviruses, including the tick-borne flaviviruses DTV and POWV. This mutation in JEV is associated with viral attenuation. This sentence is referring to the fact that, in our libraries, we did not recover variants with mutations at these positions, in contrast to D2-FL with variants at N103 and G106, indicating less mutational tolerance. However, we want to re-direct the focus of this manuscript to engineer a viable DENV that is antigenically different in the FL epitope, but not which residue is more tolerance for mutation.

1. Figure 2c: The FLM sample in the western blot shows hardly any E protein, making E/prM quantitation unreliable.

The samples used in Figure 2C derive from the growth curve endpoint (Figure 2A), in which there is a 1-log difference in viral titer between D2 and D2-FLM. Equivalent volumes of viral supernatant were loaded in the gel, explaining the reduced intensity of the E band in D2-FLM. The higher exposure on the right shows the E band more clearly for D2-FLM. The Western blot assay comparing prM/E ratio as a measure of maturation state was described and validated in our previous study (Tse et al. 2022, mbio). The methods and figure legend have been updated to include greater detail. The polyclonal E antibody was specifically chosen for this study as our previously used monoclonal antibody targeted the fusion loop. The polyclonal antibody was raised against a fragment of E (AA 1-495) and should have minimal effect by the fusion loop mutations.

1. Lines 149 -151: "Importantly, D2-FL and D2-FLM were resistant to antibodies targeting the fusion loop. While neutralization by 1M7 is reduced by ~2-logs, no neutralization was observed for 1N5, 1L6, and 4G2 for either variant (Figure 3 A)".a) Partial neutralization was observed for 1N5, for D2-FL.

The text has been updated to more accurately describe the 1N5 neutralization data.

b) Do these mAbs cover the full spectrum of fusion loop antibodies identified thus far in the field?

We did not test every known fusion loop antibody that has been described, instead focusing on 1M7, 1N5, 1L6, and 4G2, which were previously described by Smith et al and Crill et al. We also modified the text in discussion to reflect the possibility of other FL-Ab that are not affected by out mutations.

“We have tested a panel of FL-Ab; however, we cannot exclude the possibility that other FL-Abs may not be affected by N103S and G106L. However, we have shown that saturation mutagenesis could generate mutants with multiple amino acid changes, and we are currently using D2-FLM as backbone to iteratively evolve additional mutations in FL to further deviate the FL antigenic epitope.”

c) Are the epitopes known for these mAbs? It would be useful to discuss how the epitope of 1M7 differs from the other mAbs? What are the critical residues?

Critical residues for these antibodies have been described. They are as follows: 1M7: W101R, W101C, G111R; 1N5: W101R, L107P, L107R, G111R; 1L6: G100A, W101A, F108A; 4G2: G104H, G106Q, L107K. The critical residues for 1M7 are slightly different than the others, perhaps explaining the residual binding to D2-FL. Note that the critical residue identified previously for 1M7 and 1N5 do not overlap with D2-FLM mutations, suggesting the FL mutations has extending effect on the antigenic FL epitope.

d) Maybe the D2-FL mutant can be further evolved with selection pressure with fusion loop mAbs 1M7 +/-1N5 and/or other fusion loop mAbs.

We agree that it may be possible to further evolve D2-FL using antibody selection, although we have not yet performed these experiments, we are currently performing iterative saturation mutagenesis and directed evolution to further evolve away from the natural FL.

1. It would have been useful to include D2-M for comparison (with evolved furin cleavage sequence but no fusion loop mutations).

Neutralization data for some of the mAbs against D2-M can be found in our previous study (Tse et al. 2022 mBio), in which no difference in neutralization was observed compared to DV2 wildtype. Given the limited resources of the anti-DENV NHP and human serum, we did not add D2-M for comparison. Although some insight can be deduced from the D2-FL vs D2-FLM comparison, we agree future studies that are designed to delineate CR-Ab population between prM, FL and other CR-epitopes should include D2-M for comparison.

1. Data for polyclonal serum can be better discussed. Table 1 is not discussed much in the text. For the R1160-90dpi-DENV4 sample, D2-FL and D2-FLM are neutralized better than wild type DENV2? The authors' interpretation in lines 181-182 is inconsistent with the data presented in Figure 3C, which suggests that over time, there is INCREASED (not waning) dependence on FL- and prM-specific antibodies for heterotypic neutralization.

We remade Table 1 to show dilution factors instead of dilution factor-1 of FRNT_50_.

In general, our human convalescent sera from heterotypic infection (DENV1, 3 and 4) showed none to low neutralization against our DENV2. FRNT_50_s were between 1: 40 – 1:200. Given the weak potency of the antiserum, it is difficult to compare the FRNT_50_s between DV2-WT and D2-FLM.

Similarly, in a different NHP cohort (2nd NHP cohort shown in Table 1), only one DENV4 infected NHP (R1160) showed a low heterotypic titer against DENV2. The detectable FRNT_50_s were between 1: 50 – 1:90. The value was extrapolated based on a single data point (1:40) which has above 50% neutralization. Given the Hill slope of all the neutralization curves were below 0.5, readers should be cautious when analyzing these FRNT_50_ values. * denotes the Hill slopes of all the neutralization curve of them sample are <0.5 (low confidence of neutralization) in Table 1.

In conclusion, we do not think serum from Table 1 is potent enough to shows difference between the viruses. The intension to show the negative data in Table 1 is to highlight the difference in serum heterogeneity in DENV infected patients and experimental infected NHPs.

As the reviewer pointed out, the dependence of FL-Ab in later time points increased (the difference between DV2 and D2-FL at 20dpi vs 60dpi vs 90dpi), suggesting non-FL CR-Ab is waning but not prM- and FL-Abs. We rewrote the sentence as follow:

“These data suggest that after a single infection, many of the CR Ab responses target prM and the FL and the reliance on these Abs for heterotypic neutralization increase overtime (Figure 3C).”

Suggestions for further experiments-1. It would be interesting to see the phenotype of single mutants N103S and G106L, relative to double mutant N103S/G106L (D2-FL).1. The fusion capability of these viruses can be gauged using liposome fusion assay under different pH conditions and different lipids.1. Correlative antibody binding vs neutralization data would be useful.

We thank the reviewer for the suggestions; we agree these would be of interest and, indeed, these studies are currently underway. In regard to single mutants, these were present in the initial plasmid library but did not enrich after viral production and passage. Two possible explanations can be drawn, (1) The stochastic of directed evolution prevents a single mutant with similar fitness to enriched. (2) The two mutations are compensatory to each other to make a functional mutant. The 2nd hypothesis highlights the difference between saturation mutagenesis (this study) and DMS (in previous studies).

Fusion capability is indeed very interesting, however, the mechanistic difference or not between wildtype FL and the mutated FL in supporting fusion is not the focus of this study. Instead, we are currently working on adapting the D2-FLM in mammalian cells. If successful, the difference in fusion mechanism between the Vero adapted and D2-FLM in different lipid, insect vs mammalian would be of interest.

We are currently developing whole virus ELISA; we avoid using rE monomer for the study as it might neglect the conformation Ab.

**Reviewer #2 (Public Review):**
Antibody-dependent enhancement (ADE) of Dengue is largely driven by cross-reactive antibodies that target the DENV fusion loop or pre-membrane protein. Screening polyclonal sera for antibodies that bind to these cross-reactive epitopes could increase the successful implementation of a safe DENV vaccine that does not lead to ADE. However, there are few reliable tools to rapidly assess the polyclonal sera for epitope targets and ADE potential. Here the authors develop a live viral tool to rapidly screen polyclonal sera for binding to fusion loop and pre-membrane epitopes. The authors performed a deep mutational scan for viable viruses with mutations in the fusion loop (FL). The authors identified two mutations functionally tolerable in insect C6/36 cells, but lead to defective replication in mammalian Vero cells. These mutant viruses, D2-FL and D2-FLM, were tested for epitope presentation with a panel of monoclonal antibodies and polyclonal sera. The D2-FL and D2-FLM viruses were not neutralized by FL-specific monoclonal antibodies demonstrating that the FL epitope has been ablated. However, neutralization data with polyclonal sera is contradictory to the claim that cross-reactive antibody responses targeting the pre-membrane and the FL epitopes wane over time.Overall, the central conclusion that the engineered viruses can predict epitopes targeted by antibodies is supported by the data and the D2-FL and D2-FLM viruses represent a valuable tool to the DENV research community.
**Reviewer #1 (Recommendations For The Authors):**
1. Line 51-52: "Currently, there is a single approved DENV vaccine, Dengvaxia."Line 56-57: "Other DENV vaccines have been tested or are currently undergoing clinical trial, but thus far none have been approved for use."It should be specified for the global audience that this applies to the United States. Takeda's DENV vaccine, QDENGA is approved in Indonesia, European Union, and Brazil.

The text has been modified to include this information.

1. Line 62-63: - "The core fusion loop-motif DRGWGNGCGLFGK is highly conserved..."Lines 78-80: - We generated two different saturation mutagenesis libraries, each with 5 randomized amino acids: DRGXGXGXXXFGK (Library 1) and 79 DRGXXXXXGLFGK (Library 2).It may be useful for the readers if the amino acid numbers are stated.The core fusion loop motif DRGWGNGCGLFGK (Eaa98-110) is highly conserved.We generated two different saturation mutagenesis libraries, each with 5 randomized amino acids: DRGXGXGXXXFGK (Library 1; Xaa 101,103, 105-7) and DRGXXXXXGLFGK (Library 2; Xaa 101-105).

This information has been added to the text.

1. Line 91-92: "Bulk Sanger sequencing revealed an additional Env-91 T171A mutation outside of the fusion-loop region."It looks like the mutation T171A is in domain I of the E protein and does not seem to interface with the fusion loop. Is that why it wasn't pursued further?

The E171A mutation was included in the infectious clone for D2-FL and D2-FLM. The text has been modified to clarify this inclusion.

1. Lines 82-85: "Saturation mutagenesis plasmid libraries were used to produce viral libraries in either C6/36 (Aedes albopictus mosquito) or Vero 81 (African green monkey) cells and passaged three times in their respective cell types."a) What was the size of the libraries? How does one make sure that the experimental library actually has all the amino acid combinations that were intended?

Each library has 5 randomized amino acids, so there are 205 = 3.2 million combinations. In these experiments, sequencing of the plasmid libraries revealed about 2 million unique amino acid sequences, or approximately 62.5% library coverage. The actual plasmid diversity is expected to be higher than 2 million as our deep sequencing has limited coverage.

b) The wild type sequence was excluded from the libraries, correct?

The wild-type sequence was not specifically excluded from the libraries, as there is no easy method to do so. Wild-type sequence was detected in the plasmid libraries but was not selected in the C6/36 library. However, in the Vero library, we recovered WT virus.

1. Table 1: - Please include in the table description, what the colors indicate.

We remade Table 1 to show dilution factors instead of dilution factor-1 of FRNT50 and removed the unnecessary color code. We also added all relevant information in the table legend.

1. Lines 246-248: "Previously, a single study on WNV successfully generated a viable virus with a single mutation at the fusion loop, although it severely attenuated neurovirulence."It may be worthwhile to mention the WNV mutation (L107F) as some readers may be curious about where this mutation is relative to the ones described in this study.

This information has been added to the text. We also included the previously described FL mutations in flaviviruses in the text.

**Reviewer #2 (Recommendations For The Authors):**
Major Critique:There is a disconnect between Fig 2A and 2C. FL and FLM viruses have much lower levels of prM-E expression in the viral supernatants based on the western blot in 2C. Why isn't E being detected in the Western? Is the particle-to-pfu ratio skewed in the mutant viruses? Is it possible that the polyclonal is targeting the cross-reactive prM and FL epitopes, and if so would using a monoclonal antibody targeting a known DIII-epitope (2D22) yield a different western result? Also, the legend and methods for Fig 2C are not clear. What is actually being tested in the Western blot? Were equivalent volumes of the different viral preps used?

The samples used in Figure 2C derive from the growth curve endpoint (Figure 2A), in which there is a 1-log difference in viral titer between D2 and D2-FLM. Equivalent volumes of viral supernatant were loaded in the gel, explaining the reduced intensity of the E band in D2-FLM. The higher exposure on the right shows the E band more clearly for D2-FLM. The Western blot assay comparing prM/E ratio as a measure of maturation state was described and validated in our previous study (Tse et al. 2022, mBio) and the methods have been updated to include greater detail. The polyclonal E antibody was specifically chosen for this study as our previously used monoclonal antibody targeted the fusion loop. The polyclonal antibody was raised against a fragment of E (AA 1-495) and should not be affected by the fusion loop mutations. 2D22 is a conformational antibody and does not work in western blot.

Table 1: The data within Table 1 is ignored in the text, and some of this data contradicts the central conclusions of the manuscript.o A. Some of the convalescent data contradicts the hypothesis. DS0275 had an equivalent neut between DV2 and D2-FLM, DS1660, and R1160 (90) had better neut against the D2-FLM than DV2. Discussion of these samples is warranted.o C. The description in the legend does not adequately describe the table. What do the colors represent? What are the numerical values being displayed? What is in parentheses, (I assume the challenge strain)? The limit of detection is reported as 1:40; 0.25. 1:40 is 0.025 which matches most of the data? There is inadequate description of these experiments in the materials and methods.

We remade Table 1 to show dilution factors instead of dilution factor-1 of FRNT_50_ and removed the unnecessary color code. We also added discussion for Table 1 and clarify the difference between the three cohorts of serum in the text with the corresponding references.

In general, our human convalescent sera from heterotypic infection (DENV1, 3 and 4) showed none to low neutralization against our DENV2. FRNT_50_s were between 1: 40 – 1:200. Given the weak potency of the antiserum, it is difficult to compare the FRNT_50_s between DV2-WT and D2-FLM.

Similarly, in a different NHP cohort (2nd NHP cohort shown in Table 1), only one DENV4 infected NHP (R1160) showed a low heterotypic titer against DENV2. The detectable FRNT_50_s were between 1: 50 – 1:90. The value was extrapolated based on a single data point (1:40) which was above 50% neutralization. Given the Hill slope of all the neutralization curves were below 0.5, the FRNT_50_ values are not reliable.

In conclusion, we do not think sera from Table 1 is potent enough to show difference between the viruses. The intension to show the negative data in Table 1 is to highlight the difference in serum heterogeneity in DENV infected patients and experimental infected NHPs.

Minor critique:Figure 1C: Legend is not clear for this panel. What is on the x-axis of the bubble plots? Are these mutations across the entire viral genome or is this just the prM-E sequence?

The X-axis is a scatter of all of the sequences contained in the library, similar to graphs used for plotting CRISPR screen results. These represent individual sequences from the saturation mutagenesis libraries in the fusion loop of E as described in Figure 1B.

The wording in Lines 92-94 is not clear. It looks like the T171A mutation was present in 95% of the sequences (Line 92). Yet this sequence was not incorporated into the variant virus. What is the rationale for omitting this mutation in downstream variant virus generation?

The 95% in Line 92 refers to the variant containing N103S/G106L mutations as seen in Figure 1C. The high-throughput sequencing approach did not include residue 171, so the presence of the T171A mutation in combination with fusion loop mutations cannot be determined. However, the E171A mutation was included in the infectious clone for D2-FL and D2-FLM. The text has been modified to clarify this confusion.

The authors discuss the potential of the D2-FL or D2-FLM virus as a potential vaccine platform in the abstract, introduction, and conclusion. This is a good idea, but the authors provide no evidence of feasibility in this manuscript.

The ultimate goal to engineer a viable DENV with distinct FL antigenic epitope is for it use as a cost-effective, live attenuated vaccine. As this is the rationale for the study, we introduce the concept throughout the manuscript. The current study demonstrated the possibility to mutate a novel fusion loop motif in DENV and provided evidence to show the favorable antigenic properties of D2-FLM. We agree with the reviewer that definitive work in animal to show vaccine efficacy need to be done and are currently undergoing. To avoid misleading our audience, we tone down the emphasis of vaccine use in the text.

Line 150-153: Figure 3A demonstrates that the FL-specific antibodies broadly do not neutralize the mutant viruses. However, the conclusions are overstated in the text. 1N5 neutralizes the D2-FL variant.

The text has been updated to more accurately describe the 1N5 neutralization data.

Lines 175-182: The authors make a lot of assumptions about the target of the polyclonal target without any evidence.

These lines reference studies that showed greater enhancement by antibodies targeting the fusion loop and prM as compared to other cross-reacting antibodies. The assumption that both our manuscript and others have drawn was that Abs that are cross-reactive and weakly neutralizing are more prone for ADE. As discussed, other groups have attempted to mutate the FL from recombinant E protein to achieve similar goal to remove the fusion loop epitope to reduce ADE. We have re-written the sentence in the followings:

“As FL and prM targeting Abs are the major species demonstrated to cause ADE in vitro, we and others hypothesized these Abs are responsible for ADE-driven negative outcomes after primary infection and vaccination,10–12,32 we propose that genetic ablation of the FL and prM epitopes in vaccine strains will minimize the production of these subclasses of Abs responsible for undesirable vaccine responses. Indeed, covalently locked E-dimers and E-dimers with FL mutations have been engineered as potential subunit vaccines that reduce the availability of the FL, thereby reducing the production of FL Abs.33–36”